# Coronary Atherosclerosis Imaging

**DOI:** 10.3390/diagnostics10020065

**Published:** 2020-01-24

**Authors:** Michael Y. Henein, Sergio Vancheri, Gani Bajraktari, Federico Vancheri

**Affiliations:** 1Institute of Public Health and Clinical Medicine, Umea University, SE-90187 Umea, Sweden; michael.henein@umu.se (M.Y.H.); gani.bajraktari@umu.se (G.B.); 2Departments of Fluid Mechanics, Brunel University, Middlesex, London UB8 3PH, UK; 3Molecular and Nuclear Research Institute, St George’s University, London SW17 0RE, UK; 4Radiology Department, I.R.C.C.S. Policlinico San Matteo, 27100 Pavia, Italy; sergiovancheri@gmail.com; 5Medical Faculty, University of Prishtina, 10000 Prishtina, Kosovo; 6Clinic of Cardiology, University Clinical Centre of Kosova, 10000 Prishtina, Kosovo; 7Internal Medicine, S.Elia Hospital, 93100 Caltanissetta, Italy

**Keywords:** coronary atherosclerosis, coronary plaque, coronary imaging

## Abstract

Identifying patients at increased risk of coronary artery disease, before the atherosclerotic complications become clinically evident, is the aim of cardiovascular prevention. Imaging techniques provide direct assessment of coronary atherosclerotic burden and pathological characteristics of atherosclerotic lesions which may predict the progression of disease. Atherosclerosis imaging has been traditionally based on the evaluation of coronary luminal narrowing and stenosis. However, the degree of arterial obstruction is a poor predictor of subsequent acute events. More recent techniques focus on the high-resolution visualization of the arterial wall and the coronary plaques. Most acute coronary events are triggered by plaque rupture or erosion. Hence, atherosclerotic plaque imaging has generally focused on the detection of vulnerable plaque prone to rupture. However, atherosclerosis is a dynamic process and the plaque morphology and composition may change over time. Most vulnerable plaques undergo progressive transformation from high-risk to more stable and heavily calcified lesions, while others undergo subclinical rupture and healing. Although extensive plaque calcification is often associated with stable atherosclerosis, the extent of coronary artery calcification strongly correlates with the degree of atherosclerosis and with the rate of future cardiac events. Inflammation has a central role in atherogenesis, from plaque formation to rupture, hence in the development of acute coronary events. Morphologic plaque assessment, both invasive and non-invasive, gives limited information as to the current activity of the atherosclerotic disease. The addition of nuclear imaging, based on radioactive tracers targeted to the inflammatory components of the plaques, provides a highly sensitive assessment of coronary disease activity, thus distinguishing those patients who have stable disease from those with active plaque inflammation.

## 1. Introduction

Atherosclerosis is the primary cause of coronary artery disease (CAD) [1,2]. The underlying pathophysiologic mechanisms develop and progress for decades before the disease becomes clinically evident. Traditionally, atherosclerosis imaging has focused on the assessment of arterial luminal narrowing and occlusion. However, most acute coronary events occur from atherosclerotic plaque rupture or erosion causing arterial thrombosis [3]. The aim of performing imaging diagnostic tests is the assessment of patients at risk of acute coronary events associated with plaque formation, before atherosclerotic complications occur. This objective is based on the identification of coronary atherosclerotic burden, its extent and pathological characteristics which are closely associated with progression and rupture of vulnerable plaques.

The choice of imaging modality depends on the cardiovascular risk level of the patient. Non-invasive imaging is best suited in primary prevention for low- intermediate- risk population, in order to improve risk stratification and identify individuals who may benefit from intensive treatment. In high-risk patients, intravascular imaging provides accurate assessment of vulnerable plaques and early stage of their development.

## 2. Pathology

Atherosclerosis is a dynamic process strongly associated with inflammation, involving endothelial dysfunction, macrophages and vascular smooth muscle cells proliferation and apoptosis, calcification of the intima and osteogenesis [4,5,6,7,8]. Extensive imaging and postmortem histological studies have shown the essential features of coronary plaques which may progress to rupture [9,10,11,12,13,14,15]. These features include: (1) large infiltration of macrophages and to a lesser extent of T-lymphocytes, (2) a thin-cap fibro-atheroma (TCFA), (3) microcalcifications, (4) coronary artery outward or compensatory remodeling and negative remodeling (arterial shrinkage) which is associated with higher degree of coronary stenosis, and 5) plaque angiogenesis and intra-plaque hemorrhage.

Inflammation has a central role in atherogenesis, from plaque formation to rupture. The earliest lesion is increased endothelial permeability to small and dense LDL-cholesterol particles which are produced in condition of inflammation [11,16]. Endothelial retained lipoproteins promote local inflammation which exerts several pathologic effects: promoting migration of macrophages which catabolize the lipoprotein (foam cells), finally undergoing apoptosis, forming the lipid-rich necrotic core, and stimulating the proliferation and migration of vascular smooth cells into the intima, leading to the formation of fibro-atheroma [4]. This is a stable plaque consisting of a necrotic core covered by a thick fibrous cap, developed in response to endothelial damage. If the inflammation persists, the catabolic effect of macrophages, through the release of matrix-degrading metalloproteinases, results in dissolution of collagen and thinning of the fibrous cap, producing a TCFA [17,18]. This consists of a large lipid-rich necrotic core (>40% of plaque volume) separated from the coronary lumen by a thin fibrous cap, less than 65 microns thick [19,20,21]. These pathological changes make the plaque unstable [22].

Also, inflammation stimulates calcification as a healing response to the necrotic lesion. Calcification originates as aggregation of small crystals of hydroxyapatite giving rise to microcalcification, less than 50 microns in diameter, embedded in the fibrous cap. Some of these calcifications aggregate into larger masses to form spotty calcification, 1 to 3 mm in diameter. Along with TCFA, large necrotic lipid core and macrophage infiltration, microcalcifications make the plaque unstable. Their effect on plaque vulnerability is due to the local mechanical stress produced within the fibrous cap [23,24,25]. Progressive calcification promotes the transition from early stage of high-risk microcalcifications to the stable end-stage macroscopic calcifications which make the plaque stable, limit the spread of inflammation and rarely result in rupture [26]. As the plaque grows, the coronary size increases in an outward direction, thereby maintaining both the luminal diameter and the blood flow. Such positive or compensatory remodeling allows even large plaques to be accommodated without producing symptoms. This is also the reason why coronary angiography may underestimate the plaque burden since it opacifies only the arterial lumen.

Most commonly, acute coronary events are triggered by TCFA rupture exposing the underlying necrotic core to the circulating blood. This activates platelets aggregation and the coagulation cascade causing arterial thrombosis [3,27,28]. Alternatively, in about one-third of acute events, plaque erosion of the endothelium overlying the fibrous cap may lead to thrombosis [11,29,30].

Although atherosclerotic plaque imaging has traditionally focused on the detection of single vulnerable or unstable plaques, it is now well established that the atherosclerotic lesions have dynamic features and their morphology may change over few months [31,32]. This is confirmed by the observation that patients with CAD have plaques in various stages of development [33]. Although postmortem studies suggest that TCFA are most often found in the culprit lesions in acute coronary syndrome, the majority of vulnerable plaques undergo progressive transformation from high-risk lesions to more stable plaques with extensive calcification, while others undergo subclinical rupture and healing [34,35,36]. As a consequence, since most plaque ruptures are silent, the few that are responsible for the acute events cannot be distinguished from the others. Although high-risk plaques do not strictly identify future culprit lesions, they are expression of diffuse and severe atherosclerotic disease, and propensity to develop plaques with unstable features. In addition, plaque rupture may occur at multiple distant coronary sites [37,38]. These observations indicate that the global risk of the patient is more strongly associated with the extent of coronary disease and the presence of systemic factors (vulnerable patient) than with individual lesions (vulnerable plaque) [39,40,41,42,43]. Accordingly, in recent years the main target of imaging has shifted from the detection of individual coronary plaques at high-risk of rupture to the assessment of disease activity that may be more closely related to the vulnerable patient.

## 3. Non-invasive imaging

### 3.1. Plaque Morphology

Non-invasive visualization of the morphology of coronary atherosclerotic lesions is performed using computed tomography (CT), computed tomography coronary angiography (CTCA), and cardiac magnetic resonance (CMR).

### 3.2. Computed Tomography (CT)

Non-contrast electrocardiographic (ECG)-gated multidetector CT provides a direct quantitative assessment of coronary artery calcium (CAC) [44]. (Figure 1). The standard method for CAC quantification is the Agatston score which measures the density of calcification in each coronary segment, multiplied by the area and summed for all arteries [45]. CAC score correlates with the total coronary plaque burden and is an independent predictor for CV events irrespective of ages [46,47,48,49,50,51,52]. Extensive calcification is associated with higher risk of CAD, because the presence of more plaques increases the chance that one may rupture. Current guidelines recommend CAC score assessment in individuals at low- to intermediate-risk in whom treatment decision may be improved by CV risk stratification [53,54,55,56]. In individuals without known atherosclerotic disease, a CAC score of zero is associated with a high negative predictive value in excluding significant CAD [57]. However, although only 5% of individuals with zero CAC have significant stenosis, low CAC scores do not exclude obstructive CAD [57,58,59]. Instead, a CAC score >100 is associated with a risk of events similar to patients with previous CAD [60,61]. 

The relevance of CAC in the prediction of acute events is also demonstrated by the observation that individuals free of clinical CAD and without conventional risk factors but elevated CAC have substantially higher rates of all-cause mortality than those who have multiple risk factors but no CAC [62]. Also, combining CAC score and conventional risk factors carries significant sensitivity for prediction of >50% coronary stenosis which is higher than luminal stenosis measured by CTCA [48]. The relationship of the Agatston score with subsequent coronary events appear to be mainly based on the volume component of the CAC score, while calcification density has an inverse relationship with risk of CAD [63,64,65,66]. Since CAC density refers to the concentration of calcium in the plaques, this inverse relationship may reflect the protective role of stable macrocalcifications. Serial coronary CT for the assessment of the effects of treatment is now feasible due to rapid advances in image reconstruction that have substantially reduced the radiation dose without compromising the imaging quality [67]. Although calcified plaque burden assessment is important in identifying the subclinical phase of CAD, CT calcium scoring does not quantify the burden of non-calcified and unstable plaques nor coronary stenosis.

### 3.3. Computed Tomography Coronary Angiography (CTCA)

The intravenous administration of iodine contrast media allows the assessment of coronary stenosis and several adverse coronary plaque characteristics, such as spotty calcification, positive remodeling, and low-attenuation non-calcified plaques which identify a large necrotic core [68,69]. (Figure 2 and Figure 3). However, smaller components such as microcalcifications and TCFA cannot be detected because their dimension is ten times lower than the spatial resolution of CTCA (about 500 microns) [70].

CTCA is mainly appropriate for symptomatic moderate-risk patients without known CAD [71]. A negative CTCA is associated with very low risk of coronary events (negative predictive value about 99%) [72,73,74]. Hence, the test is appropriate to rule-out significant CAD, thus reducing the need for invasive tests. Compared to intravascular ultrasound (IVUS), CTCA has excellent correlation for coronary luminal and plaque area [75,76]. The prognostic value of obstructive compared to non-obstructive CAD has been assessed with CTCA, showing that it is the extent of disease, regardless of whether obstructive or non-obstructive, that provides additional prognostic value [77]. Indeed, patients with non-obstructive CAD who had extensive disease, showed similar rates of acute events as those with obstructive but less extensive disease.

Compared to conventional angiography, CTCA has been reported to have higher accuracy in identifying calcified and non-calcified plaques, and coronary positive remodeling [78,79]. While more than 95% of arterial stenosis documented with angiography are confirmed by CTCA, only one-third of those detected with CTCA are identified by angiography.

The relationship between morphological plaque characteristics and patient clinical presentation has been investigated with CTCA in three population groups with different cardiovascular risk [80]. Plaque volumes and the proportion of necrotic core progressively increased with worsening risk profile, while the proportion of densely calcified plaques reduced. These observations seem to confirm that individuals with high clinical risk profile are associated with potentially unstable lesions, while those with low-risk are associated with greater proportion of calcified and more stable plaques [68].

### 3.4. Cardiac Magnetic Resonance (CMR)

CMR provides non-invasive accurate soft tissue contrast imaging, visualization of coronary lumen and arterial wall, atherosclerotic disease burden, plaque composition and activity [81,82]. Coronary atherosclerosis characterization by CMR is based on electromagnetic signal intensity from protons in free water, triglycerides and free fatty acids in a strong magnetic field. Morphologic appearance of the atherosclerotic lesions depends on their free water concentration. Because calcification does not contain free water, densely calcified plaques appear as dark region on CMR. In contrast, plaques with low CT density, usually associated with vulnerable features, appear with high intensity on MR.

Non-contrast enhanced CMR can identify some vulnerable plaque characteristics, such as vascular remodeling, inflammation, and intra-plaque hemorrhage due to high T1 weighted signal which is associated with methaemoglobin, a key constituent of acute coronary thrombus [83]. While high-intensity intracoronary signals have been associated with early thrombus formation, high-intensity intra-wall signals have been related to the presence of macrophages and lipid-rich plaques, validated by intravascular imaging [84]. Contrast enhanced CMR based on gadolinium contrast agent provides better spatial resolution (CMR angiography). Also, the accumulation of gadolinium in the arterial wall is associated with increased endothelial permeability and inflammation [85]. CMR has also been able to assess significant coronary stenosis with a high negative predictive value [86,87,88]. However, the limited spatial resolution (1.3–1.8 mm with most current techniques), may limit the diagnostic use of CMR in clinical practice, whereas it has important research interest for future application. CMR has been recently combined with positron emission tomography (PET) for the simultaneous assessment of anatomic details and disease activity [89,90]. This hybrid PET/MR system has lower level of radiation exposure compared to PET/CT, thus allowing to monitor the progression of chronic atherosclerosis over time.

## 4. Disease Activity Imaging

### Positron Emission Tomography (PET)

Although structural imaging techniques provide an assessment of the plaque burden, they give no indication as to the extent of inflammatory plaque activity. Hence, they cannot accurately distinguish between patients with stable disease from those with increased disease activity, expressed by macrophages and microcalcifications which are associated with increased risk of developing acute CV events. Nuclear imaging such as PET can visualize different components of the atherosclerotic process, thus providing a highly sensitive assessment of coronary disease activity [91]. Specific radioactively labelled tracers targeted to pathological components of the atherosclerotic process, such as macrophages (^18^F-fluorodeoxyglucose and 68-Gallium-dotatate targeting the somatostatin receptor on the surface of macrophages) and microcalcification (^18^F-sodium fluoride), accumulate at sites of increased disease activity, releasing radiation which are detected by the PET scanner [83]. However, PET imaging has limited anatomic definition and needs to be combined with an anatomic imaging modality, such as CT or MR to provide simultaneous assessment of disease activity and morphological information. Hybrid imaging systems that incorporate PET with CT or MR scanners within the same gantry, provide simultaneous imaging combining the molecular specificity of PET imaging with the anatomic and functional characterization provided by CT or MR [92,93].

^18^F-fluorodeoxyglucose (^18^F-FDG) is a glucose analogue which is extensively used for malignancy staging a marker of metabolic activity. Its uptake by macrophages has been recently used to image vascular inflammation because the glucose metabolic activity of the macrophages involved in the atherosclerotic plaque is higher than the surrounding cells [94]. While ^18^F-FDG PET has been most investigated in the carotid arteries and aorta, its use in the assessment of coronary plaque inflammation is limited by the close proximity to myocardial tissue which has a high affinity to tracer uptake due to its high glucose metabolism. This obscures the coronary visualization, thus limiting accurate plaque analysis, and require a special patient preparation to minimize the myocardial glucose uptake [81].

^18^F-sodium fluoride (^18^F-NaF) has been originally studied to identify bone metastasis and is now used to visualize coronary calcification [95]. Fluoride ions are incorporated into hydroxyapatite which is a central component of the osteogenic mineralization [91]. Coronary atherosclerosis is strongly associated with macrophages osteogenic activity in the early stages of atherosclerosis, which results in microcalcifications found in the lipid-rich necrotic core of atherosclerotic plaques. This allows ^18^F-NaF to detect active microcalcification which are beyond the resolution of CT scan. The stronger affinity of the radiotracer with newly formed hydroxyapatite compared to the old crystals makes it possible to distinguish between actively inflamed coronary calcifications from stable ones. This is confirmed by the observation that large areas of coronary calcium detected by CT scan do not show increased ^18^F-NaF uptake. Conversely, regions with absent or minimal CT calcium demonstrate intense ^18^F-NaF uptake [96]. The preferential binding of ^18^F-NaF uptake to microcalcification is also explained by high surface area of hydroxyapatite in microcalcifications compared to large macroscopic calcifications where hydroxyapatite is internalized and not available for binding [97]. This discordance between morphologic and nuclear imaging provide complementary information and may improve differentiating stable from vulnerable plaques [98,99].

## 5. Intravascular Imaging

### 5.1. Angiography

Coronary angiography is the anatomic reference standard for coronary imaging. Providing direct visualization of the degree of luminal narrowing with excellent spatial and temporal resolution, it is the imaging modality of choice for symptomatic, high-risk patients (Figure 4). Compared to intravascular ultrasound or optical coherence tomography, coronary angiography has low sensitivity. While large calcified plaques are well visualized, the sensitivity for smaller lesions is limited, due to low spatial resolution (about 1 mm) [100]. Although angiography is optimal for outlining contrast-filled coronary lumen, it cannot provide information on structures that are below the endothelium. The atherosclerotic burden, especially in the earlier stage of disease when positive coronary remodeling may allow apparently normal lumen size despite the presence of wall plaques, is underestimated. Also, the degree of anatomic stenosis is only modestly correlated with hemodynamic functional significance [101]. Since most acute coronary syndromes are not consequences of occlusion at the site of severe stenosis but result from rupture of small plaques associated with only mild to moderate stenosis, angiography is limited in predicting CV events [2,102]. 

### 5.2. Intravascular Ultrasound (IVUS) and Optical Coherence Tomography (OCT)

IVUS and OCT allow direct cross-sectional visualization of the coronary wall and are now considered the gold standard for in vivo imaging of coronary calcification [79,100,103]. IVUS imaging is based on ultrasound reflection by coronary calcification and is more sensitive and specific than angiography (Figure 5). Compared to OCT, IVUS has lower spatial resolution and greater penetration depth, thus provide good assessment of the entire arterial wall. However, IVUS cannot penetrate calcium, hence its assessment of plaque calcification is quantitatively expressed as arc (in degrees) and length. Calcified plaques appear echo-dense (hyperechoic) and brighter than the surrounding adventitia. Some grey-scale IVUS signal intensity features have been associated with histological features of plaque instability and high risk of CAD: plaque with ultrasonic attenuation (echo-attenuated plaque), associated with fibroatheroma containing large necrotic core; echo-lucent plaque, containing an intraplaque zone of absent echogenicity, correlated with small necrotic core; spotty calcification and calcified nodule, both associated with acute coronary events [104,105]. 

However, the grey-scale atheromatous plaque classification is limited by the IVUS axial and lateral resolution (about 200 microns), not sufficient to visualize some components of the vulnerable plaque, such as microcalcifications or the TCFA which are usually smaller than 60 microns. It is also limited by the inability to penetrate calcification and to assess the composition and inflammatory state of the fibrous cap [106]. To improve the IVUS analysis of atherosclerotic plaque components, computer-assisted radiofrequency analysis of the reflected ultrasound signals have been developed in recent years to visualize color-coded plaque composition, including fibroatheroma, necrotic core, TCFA and dense calcification [104,105,106]. In addition to the greyscale IVUS, the colour coded virtual histology IVUS (VH-IVUS), based on the modulation of the frequency of the backscattered ultrasound waves, provides a detailed analysis of plaque composition and has been validated in vivo and in postmortem specimens. [104,107,108,109].

OCT uses a light source in the near-infrared range and measures the time delay of optical echoes reflected by the arterial wall, providing high-resolution cross sectional images of the plaque structure [79,110]. (Figure 6). OCT has about ten times higher axial resolution than IVUS, between 10-20 microns, and lower penetration depth, making this technique most suitable for evaluation of intraluminal structures [111]. Unlike IVUS, penetration of calcification by OCT is greater than for other tissues. Therefore, calcification thickness, area and volume may be quantified [100]. High sensitivity and specificity has been demonstrated for fibrous, fibrocalcific, and lipid-rich plaques [112]. Also, OCT is considered the only imaging modality that can directly measure TCFA and quantify the presence of macrophages and cholesterol crystal in the atherosclerotic plaque [20,104,113,114,115]. OCT imaging of contemporary presence of macrophages and microcalcifications in the same plaque with reciprocal distance smaller than 1 mm (termed co-localization), has been shown to be associated with more vulnerable plaque features [116,117]. These observations seem to indicate that OCT can provide both morphological and disease activity assessment of coronary atherosclerosis. However, the usefulness of OCT is limited by the poor penetration depth (1–2 mm) and attenuation of light transmitted through blood, thrombus and a lipid or necrotic core, all of which prevent the assessment of cross-section plaque and necrotic core area [110]. 

It must be mentioned that both IVUS and OCT have some limitations in detecting vulnerable plaques and predicting the risk of future events. Integrated IVUS and OCT devices in a single catheter has been developed to improve the diagnostic accuracy for high-risk lesions. IVUS and OCT combine the transmural imaging of IVUS with high resolution imaging of OCT [118,119]. (Table 1).

### 5.3. Near-Infrared Spectroscopy (NIRS)

Cholesterol-rich lipid core plaques are thought to be strongly associated with most cases of acute events [120]. Both IVUS and OCT have limited ability to accurately identify the extent of lipid core in the plaques. NIRS can detect lipid-rich plaques on the basis of the absorption pattern of near infra-red light by cholesterol molecules. The resulting image (chemogram) appears as a colour ring ranging from red to yellow, according to the amount of cholesterol in the plaque [121,122,123]. Clinical studies have demonstrated that NIRS can detect coronary plaques at risk for future major coronary events [124,125]. Since NIRS provides only minimal anatomic visualization, it has been combined with IVUS in a single catheter so that information from the two modalities can be acquired simultaneously [118,126].

### 5.4. Risk of Complications Associated with Coronary Atherosclerosis Imaging

Coronary atherosclerosis imaging techniques, with the exception of CMR and ultrasound, are based on intracoronary or intravenous administration of iodine contrast media and use of ionizing radiation. A potentially serious complication is the contrast-induced nephropathy (CIN), defined as an acute rise in serum creatinine of ≥0.5 mg/dl (0.04 mmol/L) or 25% above the baseline value, occurring during the first 72 hours after the procedure [127]. In patients undergoing coronary angiography the incidence of CIN is between 2% and 15%, strongly related to preexisting clinical conditions such as renal insufficiency, diabetes, advanced age, extent of CAD, and congestive heart failure [128,129,130,131,132,133,134]. Although CIN is usually transient, few patients may develop persistent renal damage and an increased risk of cardiovascular events [135,136]. Compared to catheter-based intracoronary procedures, the occurrence of CIN following intravenous contrast administration during contrast-enhanced CT is much lower and virtually absent in patients with normal renal function [137,138,139].

Visualization of coronary atherosclerosis with iodine contrast media exposes the patient to X-ray radiation. In addition to angiography, CT scan may represent a health concern, particularly in younger patients who are most likely to undergo sequential imaging for measuring CAC progression. This induces cumulative radiation exposure and risk of developing radiation-related malignancy. However, in recent years, new technologies have significantly reduced the level of radiation exposure for cardiovascular imaging. The development of CT scan angiography with prospective ECG-triggering, replacing the conventional retrospective ECG-gated, allowed significant reduction in radiation level which is now <3 mSv, comparable to the 1 year background radiation [140]. (Figure 7).

Compared to angiography and CT scanning, the dose of radioactive tracer given during a PET scan is small and patients are exposed to low levels of radiation during the test. When combined with CT scan, the radiation doses are much higher. MR/PET hybrid imaging system is therefore most likely to be useful when repeated imaging are needed.

## 6. Conclusions

Identification of coronary atherosclerotic plaque burden and their composition are of great clinical significance to assess the extent of disease and the appropriate treatment. In addition to the gold standard conventional catheter based coronary angiography, recent technological developments have provided a number of non-invasive and invasive imaging techniques of varying sensitivities and specificities in detecting coronary artery stenosis and plaque characterization, with good correlation to histology. Accordingly, in recent years the main target of coronary atherosclerosis is directed more towards early assessment of disease activity and progression than only luminal stenosis quantification. CT calcium scoring and angiography techniques are the most frequently used for clinical purposes in the assessment of atherosclerotic burden and in the evaluation of patients requiring treatment for severe myocardial ischemia. The new modalities of invasive and non-invasive coronary imaging may help identifying subgroups of high-risk patients who may benefit from aggressive pharmacological or interventional treatment. Future technological developments involving intracoronary imaging and new PET radiotracers are expected to provide other modalities with higher diagnostic accuracy.

## Figures and Tables

**Figure 1 diagnostics-10-00065-f001:**
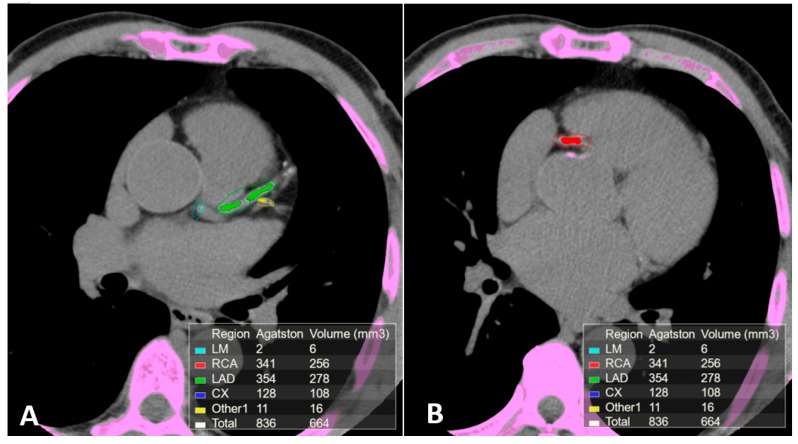
Calcium scoring (CAC) performed on a non-contrast ECG-gated CT scan of a patient with coronary arteries calcifications using commercial software Vitrea 6.0. The software automatically highlights all the structures attenuating at least 130 Hounsfield units (pink). Regions of interest (ROI) were manually placed around the coronary arteries calcifications using different colors for each coronary segment. The partial and overall Agatston score and volume of the calcifications included in the regions of interest are summarized in the box on the bottom of the images. (**A**) Calcifications in the left main, anterior descending and 1st diagonal branch. (**B**) Calcification in the proximal segment of the right coronary artery.

**Figure 2 diagnostics-10-00065-f002:**
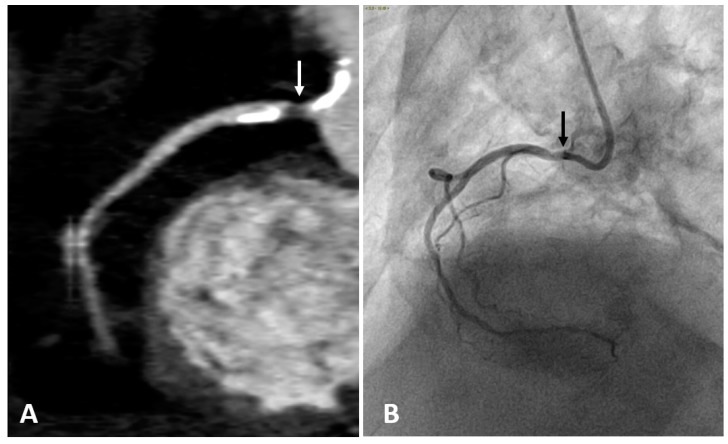
Right coronary artery (RCA) studied with CTCA (**A**) and with coronary angiography (**B**) in the same patient. Both examinations show a narrowing of the arterial lumen in the proximal RCA segment (arrows), indicating the presence of an atherosclerotic plaque. CTCA imaging shows a low-attenuation non-calcified plaque which identifies a lipid or necrotic core. Calcifications are hyperdense on CTCA, while not visible on coronary angiography.

**Figure 3 diagnostics-10-00065-f003:**
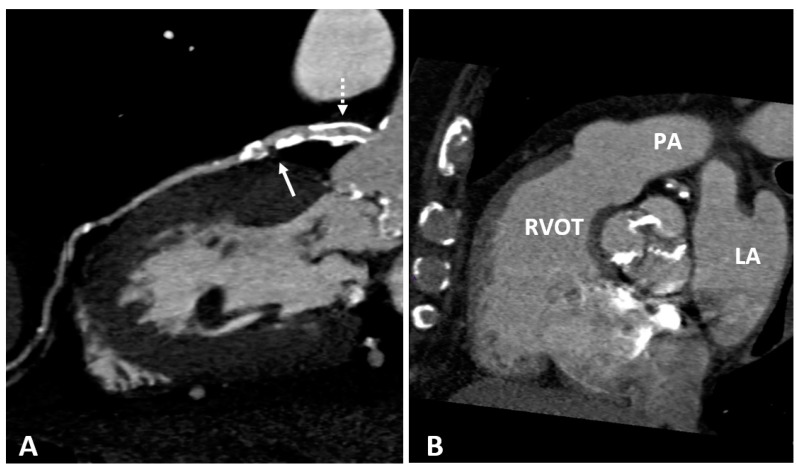
(**A**) CTCA showing the left main (LM) and left anterior descending (LAD) coronary arteries of a patient who underwent coronary angioplasty (PTCA) and stent positioning in the LM. A non-calcified low density soft plaque (white arrow) is shown between calcific plaques; the hypodense spot (dashed white arrow) within the stent may indicate initial intrastent restenosis. Calcification of the aortic valve leaflets. (**B**) CTCA of the same patient reformatted in the plane of the aortic valve shows calcification of the aortic leaflets. RVOT: right ventricle outflow tract; LA: left atrium; PA: pulmonary artery.

**Figure 4 diagnostics-10-00065-f004:**
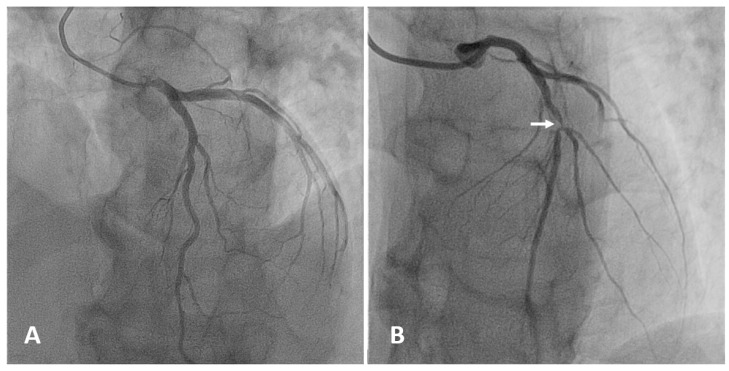
Selective angiography of the left coronary artery. (**A**) Normal coronary anatomy: left main, left anterior descending (LAD), left circumflex, and their branches are patent. (**B**) Focal severe narrowing of the lumen (arrow) due to a 90% stenosis in the proximal segment of the LAD.

**Figure 5 diagnostics-10-00065-f005:**
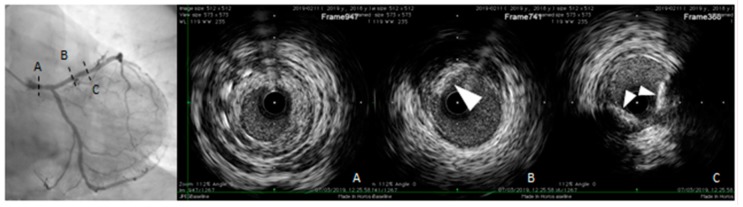
Intravascular ultrasound (IVUS) pull-back in a symptomatic, diabetic female patient with a critical left main stenosis (A, right panel: IVUS frame corresponding to the dotted line A, left panel). Left anterior descending (LAD) presents the diffuse disease with calcific eccentric lesion (triangle arrows B and C right panel, corresponding to the dotted lines B and C left panel). Behind calcifications, shadow does not allow us to obtain complete vessel contours.

**Figure 6 diagnostics-10-00065-f006:**
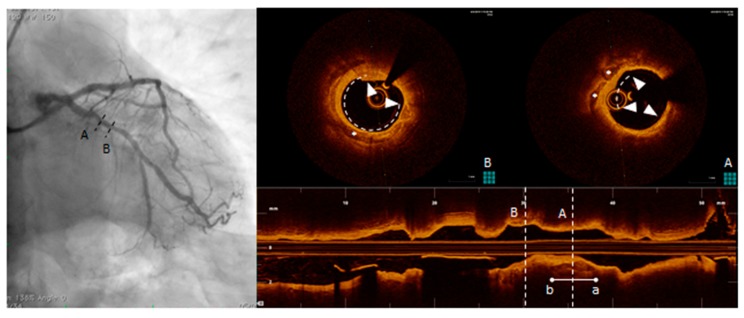
Optical coherence tomography (OCT) pull-back on left circumflex. At the proximal part of the vessel, calcific deposits were detected. From distal to proximal, calcification (triangle arrows) surround almost the whole vessel circumference (B, right panel, OCT frame corresponding to the dotted line B, left panel) and more proximally (A, right panel, OCT frame corresponding to the dotted line A, left panel) another spotty, deeper and eccentric calcification.

**Figure 7 diagnostics-10-00065-f007:**
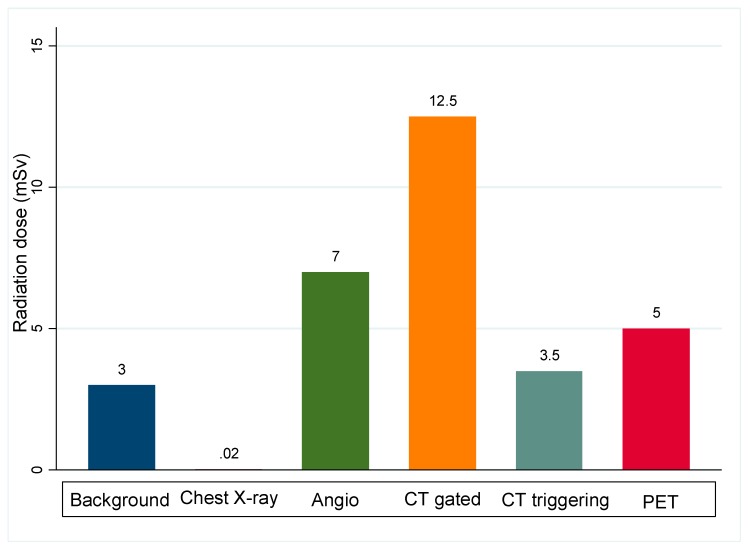
Radiation dose of coronary atherosclerosis imaging modalities, compared to natural annual background radiation dose. CT gated: Retrospective ECG-gated CT scan. CT triggering: Prospective ECG-triggering CT scan. PET: positron emission tomography.

**Table 1 diagnostics-10-00065-t001:** Properties of invasive and non-invasive imaging modalities. Cardiac magnetic resonance (CMR); positron emission tomography (PET).

	Angiography	CT	CMR	IVUS	OCT	CT/MR PET
Detection of coronary calcium	+	+++	++	+++	+++	+
Quantification of coronary calcium	+	+++	+	++	+++	+
Microcalcifications				++	++	++
Spatial resolution	1 mm	500 µm	<2 mm	200 µm	20 µm	2 µm
Depth	no limit	no limit	no limit	10 mm	2 mm	no limit
Inflammatory plaque activity			+	++	++	+++

CT Computed Tomography; CMR Cardiac Magnetic Resonance; IVUS Intravascular Ultrasound; OCT Optical Coherence Tomography; PET Positron Emission Tomography. - not possible; + poor performance; ++ reasonable; +++ good.

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
