# Peer review of "Coronary Atherosclerosis Imaging"

_diagnostics, 2020, doi:10.3390/diagnostics10020065_

Round 1

Reviewer 1 Report

This review manuscript (article) addresses the issue regarding imaging techniques for coronary arterial atherosclerosis. The authors reviewed the articles regarding CT, MRI, US for the detection of the vulnerable plaques such as thin-cap fibroatheroma, itaplaque hemorrhage, plaques with lipoprotein. The text is well written, I think this article is a very useful for the readers . But  I think that the authors should add the figures of CT, MRI or US etc. showing vulnerable plaques.

Author Response

Many thanks for your comments which can greatly improve the clarity of the manuscript.

According to your suggestion, figures of coronary angiography, CT scan, IVUS, amd OCT showing coronary stenosis and vulnerable plaques have been added

Reviewer 2 Report

Overall, good flow to review, however missing CONCLUSION section at the end - there is a paragraph that probably is your conclusion but it needs more robust details and description.  Separate section is suggested.  And maybe 1-2 paragraphs conclusion.

Section 3: Spell out the titles for the sections even though it is spelled out earlier.  Example:  3.1 Computed Tomography (CT)

Line 123, suggest "individuals free of clinical CAD"

Be careful with use of CAD vs clinical CAD.

Author Response

Many thanks for your comments.

According to your suggestion, the Conclusion section has been extended with more paragraphs and details.

Spell out the titles: done

Line 123: changed

Reviewer 3 Report

Thanks to the authors for this well written and well explaining review about the imaging techniques used for the diagnosis and follow up of the coronary atherosclerosis. The authors covered almost all the widely used imaging techniques which is nice not only for specialists but also for non specialist wanting to have a general idea about the topic. However, I still have two minor comment that might improve the manuscript quality:

-I would highly recommend the authors to provide illustration for imaging outputs in each technique or at least the main techniques in order to make it easy for the readers to pick out difference between one technique and another. A comparative summarizing table would be a plus.

-I think it would be interesting to add, when applicable, the contraindications  and side effects of the medication commonly used in coronary atherosclerosis imaging. 

Thanks 

Author Response

Many thanks for your comments.

According to your suggestion, a comparative Table summarizing the properties of invasive and non-invasive imaging modalities has been added.

A new section "Risk of complications associated with coronary atherosclerosis imaging" has been added.